# Effects of a Moderate-Intensity Interval Training Protocol on University Students’ Body Composition: A Pilot Study

**DOI:** 10.3390/jfmk10030340

**Published:** 2025-09-05

**Authors:** Bastian Parada-Flores, Luis Valenzuela-Contreras, Cristian Martínez-Salazar, Pablo Luna-Villouta, Daniela Lara-Aravena, Constanza Pino-Bárcena, Sofía Saavedra-Godoy, Álvaro Levín-Catrilao, Rodrigo Vargas-Vitoria

**Affiliations:** 1Escuela de Nutrición y Dietética, Facultad de Salud, Universidad Santo Tomás, Santiago 8370003, Chile; paradafloresbastian@gmail.com (B.P.-F.); dlara7@santotomas.cl (D.L.-A.); constanzapinob@gmail.com (C.P.-B.); saavedra.sofi16@gmail.com (S.S.-G.); 2Programa de Doctorado en Ciencias de la Actividad Física, Facultad de Ciencias de la Educación, Universidad Católica del Maule, Talca 3460000, Chile; alvarolevin7@gmail.com; 3Pedagogía en Educación Física, Universidad Católica Silva Henríquez, Santiago 8330226, Chile; lvalenz@ucsh.cl; 4Departamento de Educación Física Deporte y Recreación, Universidad de la Frontera, Temuco 4811230, Chile; cristian.martinez.s@ufrontera.cl; 5Departamento de Educación Física, Universidad de Concepción, Concepción 4070409, Chile; pabloluna@udec.cl; 6Facultad de Educación, Universidad Católica del Maule, Talca 3480112, Chile

**Keywords:** interval exercise, physical inactivity, fat mass, fat-free mass, moderate-level intensity

## Abstract

**Background**: Unhealthy lifestyles have been reported among university students, characterized by excess body weight and physical inactivity, which affects body composition and increases the risk of non-communicable diseases. Physical exercise (PE) is an effective strategy for body reshaping; however, its demands and difficulties make its practice complex. Therefore, there was an interest in investigating the effects of a low-volume, moderate-intensity interval training (MIIT) protocol. **Methods**: A 5-week quasi-experimental study was conducted. Twelve university students (ten women and two men), aged 22 ± 3.4 years, engaged in low-level physical activity. The intervention group (n = 6) performed a MIIT protocol of 1′ × 1′ × 7′ (seven series of 1 min of moderate-intensity exercise interleaved with a recovery period of 1 min) three times a week, while the control group (n = 6) continued with its regular activities. **Results**: Significant improvements were reported in body fat percentage (%BF) (*p* = 0.04; d = 0.9), fat mass (FM) (*p* = 0.00; d = 0.7) and recovery heart rate (RHR) (*p* = 0.001); d = 1) in the MIIT group compared with the pre-test and control group values. Furthermore, the body weight (*p* = 0.04) and body mass index (BMI) (*p* = 0.04) values also improved in the MIIT group, compared to the pre-test values. **Conclusions**: MIIT is an effective strategy for short-term improvement in body composition, as well as in anthropometric measures and fitness parameters, in university students with a low PAL. Its ease of applicability—based on low exercise volume and intensity—is notable, making it feasible of incorporation into student health programs.

## 1. Introduction

Excess body weight and physical inactivity are prevalent issues in the adult population, both globally [1,2] and nationally [1,3], representing two of the main behavioral risk factors for the development of non-communicable diseases, which account for over five million deaths per year worldwide [1]. In Chile, according to the latest National Health Survey [3], 74.2% of adults over 18 years of age are overweight, of whom 31.4% are classified as obese and 3.4% as morbidly obese. Additionally, it is estimated that 35.1% of these individuals have a low physical activity level (PAL).

The university stage plays a fundamental role in the development of physical activity (PA) habits and eating habits due to the social, cultural, and physiological changes students experience in their lifestyles as a result of beginning an independent and autonomous life [4]. Studies [5,6,7,8,9] report unhealthy lifestyles in Chilean university students, characterized by a low PAL, an unbalanced diet due to a low consumption of fruits, vegetables, dairy products and legumes, as well as a high intake of junk food, in addition to an excessive consumption of tobacco and alcohol [5], which adversely affects students’ health and could last once they begin their professional life, favoring the transition towards the development of non-communicable diseases [4,6,8].

A study [5] conducted on 1418 students from seven Chilean universities found that 82% of men and 95% of women revealed a low PAL, while 28% of the total sample were classified as overweight. Additionally, another study [6] that evaluated anthropometric changes and PAL in 28 university students over a four-year follow-up during their academic period reported significant increases in body weight, waist-to-hip ratio, waist circumference, and abdominal circumference (*p* < 0.01), as well as a decrease in weekly moderate-intensity PA (*p* = 0.04).

Physical inactivity represents the insufficient amount of PA performed by a person to maintain an adequate state of health, characterized by not complying with the worldwide recommendations of the World Health Organization, which indicate a minimum PA of 150 to 300 min of moderate aerobic PA a week or 75 to 150 min of vigorous PA for the adult population, which can even be combined to achieve a minimum of 600 metabolic equivalent of tasks (MET) per week [2]. Conversely, excess body weight is mainly determined by assessing the nutritional status through the body mass index (BMI) and corresponds to a BMI equal to or greater than 25 kg/m^2^ [10]. Other indicators, such as body fat percentage (%BF), can also be used, with overweight defined as BF ≥ 21% for men and ≥26% for women [11].

When PA is planned, structured, repetitive and performed for a specific objective, it is called physical exercise (PE) [12]. It has been shown to be an effective strategy for body reshaping in adults [13,14], that is, for the reduction of the fat mass and the growth of the muscle mass, resulting in significant benefits for body weight control and overall health. One of the PE methods studied is interval training, which involves a structured sequence of work and recovery intervals [15] that can be performed at high, medium or low intensity. Interval training has numerous advantages for metabolic health and improves body composition [14,16,17]. Likewise, compared to other forms of PE, such as continuous aerobic or strength training, interval training can have a similar or greater effect, leading to adaptation of the various health-related body function variables in a shorter execution time [18], and is methodologically favorable for interventions in people who are not adapted to PE and even in those with underlying pathologies [16]. In this context, the most studied and practiced modality of interval training is High-Intensity Interval Training (HIIT). However, this type of program is very demanding and difficult to execute, due to its high-effort exercises and short recovery periods, being more complex to carry out for those who are reluctant to practice PE [19]. Nevertheless, a less demanding modality of PE is moderate-intensity interval exercise (MIIT), which consists in alternating moderate- and low-intensity efforts without reaching a high intensity, while maintaining an interval-based methodological structure. In comparison with HIIT, MIIT can be regarded as a training modality that reduces workload and perceived exertion by employing lower intensity stimuli, potentially serving as an effective strategy to promote exercise participation among sedentary individuals and/or those with a low PAL. However, despite its feasibility, this approach has been barely studied within health-related research.

Considering the unhealthy habits and behaviors of the university students mentioned above [5,6,7,8,9] and the university stage being a key period for the stimulation of health promotion and prevention activities [5], the present pilot study explored an exercise protocol characterized by low exercise volume and high feasibility. The study aim was to analyze the effects of a MIIT protocol on the body composition of university students from the Maule Region, Chile. The research hypothesis stated that the 1 × 1 × 7 MIIT protocol over 5 weeks improved body reshaping in students with a low PAL.

## 2. Methods

### 2.1. Study Design

A pilot study with a quantitative approach, exploratory and descriptive nature, a longitudinal quasi-experimental design and with pre- and post-test measurements was conducted. The study included an intervention group that participated in physical exercise and a control group. Data collection took place between October and November 2024.

### 2.2. Population and Sample

A total of 12 university students, men and women aged 18–35 years, from a state-subsidized private university in the Maule Region, Chile, participated in the study. A non-probabilistic convenience sampling method was applied based on an open call. The call encompassed digital posters and email invitations.

Once the volunteers were selected, group allocation was determined according to the participants’ preferences and availability. Initially, 14 participants were recruited (7 per group); however, during the study, 1 participant withdrew due to personal reasons, and another was ruled out because of low attendance (only 50% of the exercise sessions). Thus, 12 participants remained in the intervention: 6 in the control group (5 women and 1 man) and 6 in the MIIT group (5 women and 1 man).

To minimize selection bias, identical inclusion and exclusion criteria were applied to all volunteers, and both groups underwent the same baseline evaluations once selected. All participants presented a low PAL (<600 METs/week). Their BMI ranged from 21.1 to 33.8 kg/m^2^, matching the classifications of normal weight, overweight, and obesity class I and II.

### 2.3. Inclusion Criteria

(a)Adults, both male and female, aged 18 to 35 years were included.(b)Regularly enrolled students in the university.(c)Individuals with a BMI above underweight values (≥18.5 kg/m^2^) and below grade II obesity values (<34.9 kg/m^2^).(d)Individuals with a low PAL, as reported by the International Physical Activity Questionnaire (IPAQ) [20].(e)Individuals who were physically autonomous, able to move around normally, without assistance from third parties or therapeutic instruments, and who had no medical contraindications for PE.(f)Individuals who signed the informed consent form in which they explicitly stated that their participation in the study was completely voluntary.

### 2.4. Exclusion Criteria

(a)The study ruled out individuals who could not be assessed by Bioelectrical Impedance Analysis (BIA) due to the use of permanent metallic elements, such as prostheses, pacemakers or metallic orthoses as part of a treatment.(b)Individuals who did not comply with the fasting requirement, PE restrictions, or body fluid elimination requirements indicated in the protocol for BIA.(c)Women who could not be assessed by BIA due to menstruation.(d)Individuals with musculoskeletal injuries or medical contraindications (congenital heart disease, fever, diarrhea or general malaise) that would prevent normal performance in the evaluations and interventions.(e)Individuals with permanent educational needs.(f)Pregnant women.(g)Participants attending less than 80% of the work meetings.(h)Students who terminated their relationship with the educational institution before or during the development of the project.

### 2.5. Variables

The dependent variable of this study was body composition, assessed using Bioelectrical Impedance Analysis (BIA). This test reported data on body fat percentage (%BF [%]), fat mass (FM [kg]), fat-free mass (FFM [kg]) and FFM percentage (%), as well as dry lean mass (DLM [kg]), defined as FFM without total body water.

The independent variable was the exercise intervention consisting of a MIIT protocol performed on a cycle ergometer, with a 1′ × 1′ × 7 structure. This protocol involved 7 repetitions (reps) of 1 min cycling at 70% of the heart target rate (HTR), interleaved with a 1 min active recovery period at 40% of the HTR, resulting in a total session volume of 14 min. The intervention was carried out three times a week over a period of 5 weeks. The intensity of the intervals was established according to the American College of Sports Medicine guidelines [21], alternating between moderate-intensity exercise and low-intensity recovery without complete interruption of the workload.

The secondary variables included anthropometric measures such as body weight (kg) and BMI (kg/m^2^), calculated using the formula weight (kg)/height (m^2^), and basal metabolic rate (BMR), in kcal/day. All these were estimated using the predictive equation of Bioelectrical Impedance Analysis based on sex, age, weight and height. Other secondary variables were resting heart rate (RHR [bpm]), defined as the number of beats per minute while seated at rest; THR, defined as the predetermined number of beats per minute during exercise, calculated according to Tanaka, Monahan, and Seals [22] using the formula THR = RHR + (HRmax − RHR) × percentage of intensity of PE, where HRmax = 208 − 0.7 × age [22]; and heart rate recovery (HRR [bpm]), defined as the number of beats per minute recorded one minute after the end of the exercise session, which reflects the recovery capacity of the cardiorespiratory system.

### 2.6. Materials and Instruments for Data Collection

A SECA model 769 scale with a measuring rod was used to measure the body weight (kg) and height (cm) of the participants. The data collected were used to calculate the BMI (related to one of the inclusion criteria), and the information was used for the BIA, in addition to sex and age.

The assessment of the PAL was carried out using the short version of the IPAQ [23]. This instrument consists of a self-reported questionnaire comprising seven questions regarding the physical activities performed during the previous seven days and the estimated time dedicated to them, allowing the classification of the PAL into *Low*, *Moderate* or *High* [20]. The instrument demonstrated internationally accepted validity and reliability, based on pilot tests conducted in twelve countries in the year 2000, yielding a Spearman’s correlation coefficient of 0.8 for reliability and a validity of 0.3 [23]. A short version of the questionnaire was validated and culturally adapted to the Spanish language, as well to the Chilean university context [24].

A Bodystat Quadscan 4000 BIA meter was used to evaluate the pre- and post-intervention body composition. BIA is an objective, quick, painless and non-invasive test [25], validated internationally [26], as well as in the adult population of Chile [27], which allows estimating body composition by means of the emission of a high-frequency electric current (50 kHz) that is transmitted through the body tissues by means of electrodes located at specific anatomical points, generating resistance (R) and reactance (Xc) values with which the total impedance (Z) is calculated; in addition, by means of prediction equations adjusted with data such as sex, age, height and body weight, the amounts of fat mass, fat-free mass, dry lean mass, total body water, among other indicators, are estimated [25].

During the exercise intervention, Schiller ERG 910 Plus cycloergometers were used, which have a manual load-regulation mechanism, characterized by five intensity levels.

For real-time heart rate monitoring, Polar H10 chest strap heart rate monitors connected to a wrist monitor were used. The heart rate data were recorded before, during and after each exercise session to track RHR, THR and HRR. After each interval, the evaluator instructed adjustments to the workload to maintain the heart rate within the target ranges of ±5 bpm of 70% of the THR during the work intervals and ±5 bpm of 40% of the THR during recovery.

### 2.7. Procedures

The study was divided into three phases:
Pre-test;MIIT intervention;Post-test.

#### 2.7.1. Pre-Test

All evaluations were conducted in a climate-controlled nutritional assessment laboratory at 23 °C by a professional nutritionist.

First, the body weight and height of all participants were measured to verify the inclusion criteria related to the BMI (criterion “c”). Subsequently, a Physical Education researcher assessed the PAL using the short version of the IPAQ to verify the inclusion criterion “d”.

On the same day, BIA testing was conducted. The participants were positioned in the supine position on a flat, padded wooden table to avoid interference from metallic elements. Their arms were extended and positioned approximately 30° away from the torso, and their legs were also extended, with the feet resting on the surface and slightly apart. Following the manufacturer’s protocol strictly [28], one pair of electrodes was placed on the posterior aspect of the right wrist and hand, with the positive electrode near the proximal third metacarpophalangeal joint, and the negative electrode at the ulnar head. The second pair of electrodes was placed on the right ankle and dorsum of the foot, with the positive electrode near the proximal third metatarsophalangeal joint, and the negative electrode on the lateral malleolus.

To ensure objectivity, the participants were instructed to attend the BIA session after a six-hour fast, having refrained from exercise for at least 12 h, without metallic objects, after urinating. In the case of the female participants, they were instructed to record their menstrual cycle (counting from the first day of their last menstrual period). This record had the objective to avoid BIA assessments during menstruation and to monitor potential alterations in the results associated with changes in body fluid distributions [24]. All these conditions were verified by applying a checklist prior to the measurements.

After completing the evaluations, all selected participants (intervention and control groups) were asked to keep their usual dietary habits throughout the study. No specific dietary guidelines were provided to assess the effect of the program in a real-life context. The participants were also instructed to maintain their PAL, which was verified using the post-test IPAQ.

Finally, an introductory PE session was conducted with the intervention group to explain the protocol to be followed during the MIIT sessions.

#### 2.7.2. Intervention with PE

The PE sessions consisted of a 5 min general warm-up, a 14 min MIIT execution phase and a period of gradual reduction in intensity of 5 min (cooling-down).

The warm-up included a series of low-intensity dynamic exercises aimed at cardiovascular activation, joint mobility and muscle stretching, primarily targeting the lower body segments, with mobility exercises for the ankles, knees and hips, as well as dynamic stretches for the gastrocnemius, hamstrings, quadriceps and gluteal muscles.

The exercise intervention consisted of an exploratory 5-week MIIT protocol with a 1′ × 1′ × 7 structure, performed three times per week. The protocol was designed to provide a sufficient cardiovascular and metabolic stimulus to induce initial adaptations in individuals with low PAL.

The cooling-down phase consisted of muscle stretching exercises similar to those performed during the warm-up phase, combined with respiratory controlled exercises right after each session.

#### 2.7.3. Post-Test

At the end of the MIIT program, body composition assessments were again performed, in addition to assessments of secondary variables such as body weight, BMI, BMR, PAL, RHR and HRR, using the same protocols as those of the pre-test, to compare the pre- and post-intervention results.

### 2.8. Statistical Analysis

The data were tabulated and analyzed using SPSS Statistics, version 22. The Shapiro–Wilk test was used to assess the normality of the data, revealing a normal distribution for body weight (*p* = 0.69), BMI (*p* = 0.49), FM (*p* = 0.73), BF% (*p* = 0.19), FFM (*p* = 0.35), FFM% (*p* = 0.19), RHR (*p* = 0.49), HRR (*p* = 0.11), PAL (*p* = 0.67) and BMR (*p* = 0.49), whereas age (*p* = 0.01) and DLM (*p* = 0.02) exhibited a non-normal distribution.

The baseline characteristics of the sample were compared using the independent samples *t*-test for parametric variables and the Mann–Whitney U test for non-parametric variables. Paired samples t-tests and the Wilcoxon signed-rank test were used to compare parametric and non-parametric variables, respectively, between pre- and post-intervention measurements.

A two-way repeated ANOVA measurement was applied to assess within-subject effects (pre- vs. post-intervention) and between-subject effects (MIIT vs. control group). Additionally, Cohen’s d was calculated to determine the effect size of the analyses, with the results classified as small (0.2), moderate (0.5) or large (0.8) [29]. Statistical significance was set at *p* < 0.05, with a 95% confidence interval.

## 3. Results

MIIT and control groups’ baseline characteristics did not vary significantly (Table 1). The overall session attendance rate for the total sample was 96.7% (MIIT group: 93%; control group: 100%).

In a comparison between pre- and post-intervention, the MIIT program resulted in significant improvements in body weight (*p* = 0.048), BMI (*p* = 0.046), FM (kg) (*p* = 0.047) and HRR (*p* = 0.01), indicating that the intervention positively affected indicators related to nutritional status, body composition and exercise adaptation. Additionally, the between-subject analysis revealed significant post-intervention differences in body composition between the intervention and the control groups, specifically in BF% (*p* = 0.04) and FM (*p* = 0.007), with large (d = 0.97) and moderate (d = 0.92) effect sizes, respectively, corresponding to a time × group interaction (Table 2).

HRR (*p* = 0.001) improved significantly compared to pre-intervention HRR and that in the control group, with large effect sizes (d = 1.56; d = 1.87).

Conversely, the control group exhibited a significant increase in DLM (*p* = 0.04) and FM (*p* = 0.04) when comparing pre- and post-test values.

Student’s t-test for related samples was applied to compare the pre- and post-intervention parametric variables between the MIIT group and the control group, and the Wilcoxon test was used for the non-parametric variables. Likewise, Student’s *t*-test for independent samples was applied to compare the means between the two groups (MIIT vs. control) for the parametric variables, and the Mann–Whitney U test for the same objective for the nonparametric variables. For intergroup and intragroup comparisons, the ANOVA test for repeated measures was applied. The effect size was also determined using Cohen’s d test.

## 4. Discussion

The main findings of this pilot study indicated significant improvements from the MIIT program in FM parameters, in terms of both percentage (BF%) and absolute values (kg), as well as in body weight, BMI, HRR and PAL. These results demonstrate that a low-volume, moderate-to-low-intensity interval exercise protocol is beneficial for body composition, anthropometric measures and fitness-related variables in university students with a low PAL. This highlights the importance of exercise in this population, particularly considering that reductions in body weight, BMI and FM are associated with protective effects against the risk of developing non-communicable diseases [30].

A decrease of 1 kg in FM was reported following 15 sessions of 14 min each, at an interval intensity ranging between 40 and 70% of the THR, over a 5-week period, highlighting a low weekly volume of 42 min and a total intervention volume of 210 min. In this context, a study by Viñuela et al. [31] reported an 8.1% reduction in body fat (*p* < 0.028) in young adults with a low PAL through 4 weeks of Wingate-based interval training (3 sprints of 30 s at high intensity, interleaved with 4 min of active recovery), presenting low session volume (12 sessions of 13.5 min), weekly volume (40.5 min), and total volume (162 min) [31], characteristics similar to those of the present pilot study.

Furthermore, Nalcakan et al. [32] reported a 7.3% reduction in body fat (*p* < 0.01) over 7 weeks through a sprint-based program similar to that of Viñuela et al. [31], consisting of 21 sessions and a total volume of 283 min, whereas Guzel et al. [33] reported a reduction of 2.5 kg in FM in individuals with a low PAL through low-volume moderate-intensity aerobic exercise (50–70% HRmax) over 10 weeks (weekly volume: 75–120 min).

Considering that the WHO recommendations for adults [2] suggest a minimum of 150 min of moderate-intensity aerobic activity, 75 min of vigorous activity, or a combination of them achieving an energy expenditure ≥ 600 MET·min/week to obtain health benefits, the present pilot study, along with the aforementioned studies [31,32,33], demonstrate significant effects on body composition with a lower exercise volume and a shorter duration. These results can be explained by the initial physiological and metabolic adaptations that occur rapidly in individuals with a low PAL [34], such as improvements in aerobic capacity mediated by VO2max, fat oxidation rate, basal metabolic rate, among other factors, suggesting that FM reductions are not solely dependent on energy expenditure during exercise [35].

According to Gaitán et al. [34], only 2 weeks of moderate-intensity continuous aerobic exercise (70% HRmax) or high-intensity interval exercise (50–90% HRmax) induced improvements in VO2peak (0.1 and 1.9 mL/kg/min, *p* < 0.01, respectively) and fat oxidation rate (0.3 and 1.3 mg/kg/min, *p* < 0.01, respectively) in sedentary adults, which was associated with a better body composition profile. Thus, increases in these cardiorespiratory fitness components significantly favor reductions in body fat.

Furthermore, MIIT induced significant changes in HRR, a parameter related to cardiovascular adaptations, defined as the difference between the HRmax achieved during exercise and the heart rate measured 1 min post-exercise. HRR is a determinant of recovery, and its increase is associated with improved fitness and cardiovascular health [36,37]. In the present study, HRR improved by 8,5 bpm compared to the pre-test value (*p* = 0.01) and showed a difference of 10,8 bpm relative to that in the control group (*p* = 0.001), with a large effect size (d = 1.56). This effect could reflect increased parasympathetic tone and reduced resting sympathetic activity, as well as cardiovascular adaptations resulting from short periods of regular aerobic training [36] and associated with improved cardiorespiratory fitness and body weight loss [37].

It is important to note that no studies were found in the literature specifically examining MIIT effects in university students; however, these findings can be related to other interval training programs. A study [38] conducted in sedentary adult women aged 30–45 years with overweight and obesity, aimed to compare MIIT with HIIT in relation to metabolic and body composition variables, reported significant benefits to FM for both exercise modalities without differences between groups, despite differences in intensity. However, the MIIT intervention consisted in interval running at 60–80% of heart rate reserve over 24 sessions of 30–60 min across 8 weeks, with considerably greater duration, volume and intensity than those in the present pilot study. Other interval training studies [39,40] applied low-volume approaches to improve body composition in university students; however, their methodology resembled HIIT, with intervention durations of 8–12 weeks and higher training loads, suggesting that the present pilot study offers a more time-efficient approach.

The main limitations of this study include its short duration (5 weeks), the small sample size (n = 12) and the non-probabilistic sampling, which limit its transferability. However, as a pilot study, it provides initial evidence of the effects of a scarcely explored exercise program in the health and university context. The analyses did not consider sex differences or adjustments for potential confounders related to demographics (e.g., socioeconomic status, urban/rural residence) or lifestyle factors (diet, sleep, alcohol/drug use, smoking), which are relevant to understand exercise effects on body composition holistically.

Conversely, a major strength of this study is the development of a low-impact, low-volume, short-duration exercise protocol, which is crucial to encourage participation. Given that lack of time is one of the main barriers to PA reported by Chilean university students [41], the present pilot results suggest that MIIT can be an efficient strategy to improve body composition and facilitate exercise initiation and adaptation in individuals with a low PAL.

### Guidelines for Future Research

It is recommended to implement interventions using the MIIT protocol in larger and more representative samples, ensuring a balance between sexes and applying random allocation of participants. Furthermore, it is necessary to control covariates (mentioned in the study limitations), particularly dietary intake and sleep.

Since this pilot study reports initial effects of exercise in individuals with a low PAL, it is important to compare the 1 × 1 × 7 MIIT protocol with other methodological structures or with other exercise modalities, such as continuous aerobic training, HIIT, resistance training or combined methods, in order to evaluate their impact on body composition and other metabolic health parameters, as well as the sustainability of the effects through long-term follow-up.

## 5. Conclusions

The 1 × 1 × 7 MIIT protocol is an effective strategy for improving body composition in university students with a low PAL, specifically in reducing FM, as well as improvements in body weight, BMI and HRR, achieving significant short-term benefits.

The practical applicability of the protocol is notable, given its structure based on low-volume exercise and interleaved intervals of low-to-moderate-intensity exercise, suggesting that it could be incorporated into university-level student health exercise initiation programs.

## Figures and Tables

**Table 1 jfmk-10-00340-t001:** Baseline characteristics of the participants in the intervention (MIIT) and control groups.

	MIIT Group (Media ± SD)	Control Group (Media ± SD)	*p*-Value
n	6	6	
Sex (F/M)	5/1	5/1	
Age (years old)	22.6 ± 4.8	21.3 ± 2.4	0.56
Body weight (kg)	66.7 ± 13.3	76.1 ± 16.1	0.3
BMI (kg/m^2^)	28.8 ± 8.3	31 ± 5.8	0.61
FM (kg)	24.9 ± 11.8	31.9 ± 11.8	0.66
BF% (%)	35.8 ± 11.2	40.5 ± 7.8	0.42
FFM (kg)	41.7 ± 4.2	44.1 ± 4.6	0.37
FFM% (%)	64.1 ± 11.2	59.4 ± 7.8	0.42
DLM (kg)	11.5 ± 1.3	13.5 ± 2.1	0.05
RHR (bpm)	83 ± 12.9	79.8 ± 7.8	0.65
THR 70% (bpm)	163 ± 3.8	164 ± 2.4	0.66
THR 40% (bpm)	129 ± 7.4	128 ± 5.9	0.79
RHR (bpm)	30 ± 4.5	30.1 ± 5.1	0.86
PAL (METs/week)	401 ± 119	410 ± 123	0.90
BMR (kcal/day)	1405 ± 67.5	1489 ± 111	0.14

Note. SD: standard deviation; F: female; M: male; BMI: body mass index; FM: fat mass; BF%: body fat percentage; FFM: fat-free mass; FFM%: fat-free mass percentage; DLM: dry lean mass; RHR: resting heart rate; THR: target heart rate; HRR: heart rate recovery; PAL: physical activity level; MET: metabolic equivalent of task; BMR: basal metabolic rate.

**Table 2 jfmk-10-00340-t002:** Differences in intragroup and intergroup outcome measures between MIIT and control groups.

Variables	MIIT Group				Control Group				MIIT Group vs. Control Group		
	Mean ± SD		Mean Diff. (CI; 95%)	*p*	Mean ± SD		Mean Diff. (IC; 95%)	*p*	*f*	*p*	*d*
	Pre	Post			Pre	Post					
Body weight (kg)	66.7 ± 13.3	65.7 ± 13.2	1.06 (0.03; 2.1)	0.04 *	76.1 ± 16.1	76.4 ± 16.4	−0.31 (−1.64; 1.01)	0.56	4.45	0.06	0.71
BMI (kg/m^2^)	28.8 ± 8.3	28.4 ± 8.2	0.45 (0.00; 0.89)	0.04 *	31 ± 5.8	31.1 ± 5.9	−0.11 (−0.68; 0.45)	0.62	4.04	0.07	−0.37
BF% (%)	35.8 ± 11.2	34.9 ± 3.7	0.96 (−0.43; 2.36)	0.13	40.5 ± 7.8	40.8 ± 7.7	−0.28 (−0.68; 0.12)	0.13	5.56	0.04 *	0.97
FM (kg)	24.9 ± 11.8	23.9 ± 11.3	1 (0.02; 1.97)	0.04 *	31.9 ± 11.8	32.3 ± 11.9	−0.36 (−0.72; −0.0)	0.04 *	11.3	0.00 *	0.72
FFM% (%)	64.1 ± 11.2	65 ± 10.7	0.95 (−2.31; 0.419)	0.13	59.4 ± 7.8	59.1 ± 7.7	0.28 (−0.12; 0.68)	0.13	4.92	0.05	0.63
FFM (kg)	41.7 ± 4.2	41.7 ± 3.5	0.03 (−1.17; 1.24)	0.94	44.1 ± 4.6	44.1 ± 4.8	0.05 (−0.95; 1.05)	0.90	0.01	0.97	−0.57
DLM (kg)	11.5 ± 1.3	11.4 ± 1.1	0.11 (−0.29; 0.52)	0.49	13.5 ± 2.1	13.4 ± 2.2	0.066 (0.01; 0.12)	0.04 *	0.09	0.76	1.14
RHR (bpm)	83 ± 12.9	81.3 ± 13	1.33 (−0.24; 2.91)	0.08	79.8 ± 7.8	78.8 ± 8.4	1 (−0.75; 2.75)	0.20	0.13	0.72	0.22
HRR (bpm)	30 ± 4.5	38.5 ± 7.3	−8.83 (−14.4; −3.22)	0.01 *	30.1 ± 5.1	28.1 ± 5.9	2 (−1.04; 5.04)	0.15	19.0	0.00 *	1.56
BMR (kcal/day)	1405 ± 67.5	1408 ± 60.9	−2.83 (38.2; 32.6)	0.84	1489 ± 111	1497 ± 128	−8.16 (−12.7; 5.71)	0.35	0.11	0.74	−0.88

Note. CI: 95%; *: *p* < 0.05; SD: standard deviation; f: repeated measures ANOVA; d: Cohen’s d; BMI: body mass index; BF%: body fat percentage; FM: fat mass; FFM%: fat-free mass percentage; FFM: fat-free mass; DLM: dry lean mass; RHR: resting heart rate; HRR: heart rate recovery; bpm: beats per minute; BMR: basal metabolic rate.

## Data Availability

Data is contained within the article.

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
