# Peer review of "Effects of a Moderate-Intensity Interval Training Protocol on University Students’ Body Composition: A Pilot Study"

_jfmk, 2025, doi:10.3390/jfmk10030340_

Round 1
Reviewer 1 Report
Comments and Suggestions for Authors
Dear authors
Please present the link where the study protocol was previously registered.
Present the project approval number from an ethics committee.
How participants were selected for each group.
How selection bias was avoided.
The calculated sample is incorrect, and the n is extremely small. Based on an alpha of 5% and a beta of 80%, the sample size (n) must be 68; 34 for each group.
What recommendations were given to participants regarding diet and physical activity outside the program, and how was their compliance monitored?
How was the BMR calculated?
The population characteristics are very broad.
How was it ensured that individuals attended the evaluations under similar physiological conditions: fasting, hours of sleep, hours without exercise, hydration status, etc.
According to your information, two 14-minute sessions per week with 60% attendance, the weekly exercise time was 38 min. Do you consider this protocol coherent and feasible to find a treatment effect? If so, please justify it.
According to your information, two weekly 14-minute sessions with 60% attendance are equivalent to a total weekly participation time of 18 minutes. This corresponds to a weekly caloric expenditure of less than 100 kcal. What is the scientific justification for considering that 18 minutes of weekly exercise, with a caloric expenditure of less than 100 kcal, was sufficient and appropriate?
Based on the above, do you consider the protocol coherent and feasible to find a treatment effect?
Present the average participation time of participants in the program. What is the justification for considering 60% attendance as valid? Provide a citation to justify this.
The effect size is used to determine the potency of the treatment, not the differences in baseline characteristics between populations.
The statistical analyses used are incorrect. The correct approach is a two-way repeated-measures ANOVA or a repeated-measures t-test with residual values. Furthermore, the reported p-values are incorrect. In no case were the p-values less than 0.05.
Therefore, the results, discussions, and conclusions are unfounded.
Author Response
Comment 1: Please present the link where the study protocol was previously registered.
Response 1: This pilot study was not previously registered in protocol registration platforms, such as ClinicalTrials.gov or others, due to its exploratory nature and the fact that it was part of an internal project approved by both the institution and an ethics committee (lines 441-445)
Comment 2: Present the project approval number from an ethics committee.
Response 2: The manuscript now includes this information in the “Ethical considerations” section (lines 441-442).
Comment 3: How participants were selected for each group.
Response 3: Corrected. The “Population and sample” section now specifies that a non-probabilistic convenience sampling method was used, with an open call for participation, and that groups were formed according to the volunteers’ preference and availability (lines 126-129).
Comment 4: How selection bias was avoided.
Response 4: The risk of bias was reduced by applying the same inclusion and exclusion criteria to all applicants and performing the same baseline assessments for both groups. Additionally, all participants were asked to maintain their eating habits and physical activity levels during the study (lines 135-139)
Comment 5: The calculated sample is incorrect, and the n is extremely small. Based on an alpha of 5% and a beta of 80%, the sample size (n) must be 68; 34 for each group.
Response 5: This project was designed as an exploratory pilot study, whose primary objective was to assess the feasibility and estimate the effect of the MIIT protocol for subsequent studies with larger samples. Therefore, no formal sample size calculation based on inferential hypotheses was conducted; instead, a small group of volunteer participants was used for this preliminary stage (lines 125-134).
Comment 6: What recommendations were given to participants regarding diet and physical activity outside the program, and how was their compliance monitored?
Response 6: Corrected. The “Procedures” section states that participants were asked to maintain their usual eating habits and physical activity level (PAL). No specific dietary guidelines were provided, in order to assess the effect of MIIT under real-life conditions. Compliance was monitored through the IPAQ questionnaire applied pre- and post-intervention, confirming that PAL remained low in the control group (lines 248-250).
Comment 7: How was the BMR calculated?
Response 7: Corrected. In the “Variables” section, it is specified that Basal Metabolic Rate (BMR) was calculated by the bioimpedance device based on sex, age, weight, and height, using the predictive equation built into the instrument (lines 182-183)
Comment 8: The population characteristics are very broad.
Response 8: It is acknowledged as a limitation that the sample included a wide BMI range (normal weight to class II obesity) and age range (18–35 years), which may contribute to the heterogeneity of results. This is mentioned in the discussion, and future studies are recommended to stratify or adjust the sample.
Comment 9: How was it ensured that individuals attended the evaluations under similar physiological conditions: fasting, hours of sleep, hours without exercise, hydration status, etc.
Response 9: Corrected. In the “Procedures: Pre-test” section, the evaluation instructions are described, and it is indicated that a checklist was used to verify compliance with these conditions (line 245).
Comment 10: According to your information, two 14-minute sessions per week with 60% attendance, the weekly exercise time was 38 min. Do you consider this protocol coherent and feasible to find a treatment effect? If so, please justify it.
Response 10: Corrected. The minimum attendance threshold (60%) was removed. The “Results” section now reports the average attendance for the total sample and for each group, which was 93% in the experimental group (line 294). The typographical error indicating that the MIIT protocol was conducted twice per week was corrected (line 176). Regarding volume, the low weekly volume of this protocol allowed us to assess whether a minimal stimulus could elicit initial adaptations in individuals with low PAL, which was confirmed by improvements in %BF, FM, BMI, and HRrec in only 5 weeks. Furthermore, as noted in the discussion, previous studies (Viñuela-García et al., 2016; Nalcakan et al., 2014) have reported significant improvements in body composition with protocols involving low session volume (15 min), low weekly volume (45 min), and only 4 weeks of duration (lines 347-358).
Comment 11: According to your information, two weekly 14-minute sessions with 60% attendance are equivalent to a total weekly participation time of 18 minutes. This corresponds to a weekly caloric expenditure of less than 100 kcal. What is the scientific justification for considering that 18 minutes of weekly exercise, with a caloric expenditure of less than 100 kcal, was sufficient and appropriate?
Response 11: The average attendance to the program is reported (line 294). Although this volume remains low, the justification for its application is based on prior evidence from low session and weekly volume studies and their significant effects on body composition. It is proposed that a brief, structured, and controlled stimulus can lead to initial improvements in body composition (lines 347-358).
Comment 12: Based on the above, do you consider the protocol coherent and feasible to find a treatment effect?
Response 12: The results in %BF, FM, and BMI support the feasibility of the protocol as an effective initiation dose to improve body composition in individuals with low physical activity level (PAL).
Comment 13: Present the average participation time of participants in the program. What is the justification for considering 60% attendance as valid? Provide a citation to justify this.
Response 13: Incorporated (lines 347-375).
Comment 14: The effect size is used to determine the potency of the treatment, not the differences in baseline characteristics between populations.
Response 14: Corrected. Effect size data for comparing baseline characteristics were removed (Table 1).
Comment 15: The statistical analyses used are incorrect. The correct approach is a two-way repeated-measures ANOVA or a repeated-measures t-test with residual values.
Response 15: Corrected. The wording error regarding the type of ANOVA applied has been fixed (line 287). The test applied was always the one suggested by the reviewer (two-way repeated-measures ANOVA); however, in the Methods section it had been incorrectly described.
Comment 16: Furthermore, the reported p-values are incorrect. In no case were the p-values less than 0.05.
Therefore, the results, discussions, and conclusions are unfounded.
Response 16: The statistical analysis tests were repeated using SPSS software, confirming that the p-values reported in Table 2 are correct.

Reviewer 2 Report
Comments and Suggestions for Authors
The current investigation sought to evaluate the impact of an exercise program of moderate intensity on several body composition variables as well as on nutritional as well as physical fitness indicators.
Whilst the subject matter is relevant, there are some issues with the paper that need to be discussed.
Line 34 – Only the body composition variables were identified as being selected by authors. Nonetheless, further outcomes from the study are not provided.
Line 38 – The authors should define all acronyms used.
Line 45 – The authors should include "moderate intensity" as a keyword.
Line 55 – How common is this problem in the target population (university students)? While the authors mention worldwide statistics, more precise statistics that apply directly to the population of interest would be helpful.
The research gap to be addressed in this investigation should be more explicitly stated. Explanations of what distinguishes HIIT from MIIE are subpar. Also, the reason for selecting the moderate-intensity aerobic program should be congruent with the features of the study population, with a clear structure of intervention that would make this selection reasonable.
Moreover, the hypothesis is currently unclear, especially given that the study variables are not clearly presented.
The authors would be able to specify in the methodology section what measures they used to asses each variable. In that case, a separate "variables" subsection would not be necessary.
Line 211 – The prescription of this exercise protocol should be justified.
The authors should further describe how exercise intensity was tracked. How did they ensure that individuals were retaining 70% of WHR? A bibliographic citation to justify this intensity threshold should be provided.
Line 234 – Line 234 – The authors identified that Shapiro-Wilk was use, however, the results of it were not presented.
Line 249 – Why was 60% attendance as a minimum threshold specified?
Line 252 – The phrase "physical inactive" is not adequate and should be revised.
In terms of discussion, aside from recognizing associations between variables, one should investigate potential mechanisms of association.
The difference between genders also should be considered as a limitation.
The authors should further clearly delineate between directions for future research, limitations, and conclusions in an adequately organized section.
Line 365 – “The cycle ergometer MIIE program”. It is different from the purpose of the study. However, it is more specific, and the author should consider using it earlier in the manuscript. (e.g., an aerobic moderate-intensity interval exercise program). I think that it makes more sense than a wider concept like the exercise.
Lastly, a word of appreciation to the authors for their efforts and wish that my response would contribute towards enhancing the quality of the paper.
Author Response
Comment 1: Line 34 – Only the body composition variables were identified as being selected by authors. Nonetheless, further outcomes from the study are not provided.
Response 1: Body composition is mentioned as the main variable, while throughout the text the results are presented in anthropometric and physical fitness variables as secondary effects.
Comment 2: Line 38 – The authors should define all acronyms used.
Response 2: Corrected (line 40-44).
Comment 3: Line 45 – The authors should include "moderate intensity" as a keyword.
Response 3: Corrected: the concept was incorporated into the keywords (line 49-50).
Comment 4: Line 55 – How common is this problem in the target population (university students)? While the authors mention worldwide statistics, more precise statistics that apply directly to the population of interest would be helpful.
Response 4: Corrected: statistical data from the population context were added (line 69-75).
Comment 5: The research gap to be addressed in this investigation should be more explicitly stated. Explanations of what distinguishes HIIT from MIIE are subpar. Also, the reason for selecting the moderate-intensity aerobic program should be congruent with the features of the study population, with a clear structure of intervention that would make this selection reasonable.
Response 5: Corrected: the difference between HIIT and MIIT methods and the rationale for selecting this method were further detailed (line 101-108).
Comment 6: Moreover, the hypothesis is currently unclear, especially given that the study variables are not clearly presented.
Response 6: Corrected: a research hypothesis was incorporated (lines 114-115).
Comment 7: The authors would be able to specify in the methodology section what measures they used to asses each variable. In that case, a separate "variables" subsection would not be necessary.
Response 7: We appreciate the suggestion; however, it was decided to keep the variables section, as it is considered to provide greater clarity to the study.
Comment 8: Line 211 – The prescription of this exercise protocol should be justified.
Response 8: Corrected: the requested content was incorporated (lines 177-179; 263-265).
Comment 9: The authors should further describe how exercise intensity was tracked. How did they ensure that individuals were retaining 70% of WHR? A bibliographic citation to justify this intensity threshold should be provided.
Response 9: Corrected: the way exercise intensity was monitored is detailed (lines 216-221). Its use is justified on lines 177-179.
Comment 10: Line 234 – Line 234 – The authors identified that Shapiro-Wilk was use, however, the results of it were not presented.
Response 10: Corrected: the results of the statistical test were incorporated (lines 278-281).
Comment 11: Line 249 – Why was 60% attendance as a minimum threshold specified?
Response 11: Corrected: the 60% attendance threshold was removed, as it was not selected based on a justified theory. The participants’ average attendance was detailed in the "Results" section, and since the minimum attendance was 80% (lines 294-295).
Comment 12: Line 252 – The phrase "physical inactive" is not adequate and should be revised.
Response 12: Corrected: the term "physically inactive" was replaced throughout the text with "person/individual with a low physical activity level (PAL)."
Comment 13: In terms of discussion, aside from recognizing associations between variables, one should investigate potential mechanisms of association.
Response 13: Incorporated: associations between the variables were established (lines 370-375; 380-385).
Comment 14: The difference between genders also should be considered as a limitation.
Response 14: Incorporated (lines 401-404).
Comment 15: The authors should further clearly delineate between directions for future research, limitations, and conclusions in an adequately organized section.
Response 15: Incorporated: a separate section was added (lines 398-404; 405-410; 412-422).
Comment 16: Line 365 – “The cycle ergometer MIIE program”. It is different from the purpose of the study. However, it is more specific, and the author should consider using it earlier in the manuscript. (e.g., an aerobic moderate-intensity interval exercise program). I think that it makes more sense than a wider concept like the exercise.
Response 16: The concept was removed (lines 424).

Reviewer 3 Report
Comments and Suggestions for Authors
The article “Effects of a moderate intensity interval exercise program on university students' body composition: a pilot study” addresses a relevant public health issue—physical inactivity and excess weight among university students—by evaluating a moderate-intensity interval exercise (MIIE) protocol. The concept is timely, the experimental design is broadly acceptable for a pilot study, and the reported findings are potentially valuable for health promotion in young adult populations. However, the article exhibits several major and minor issues in study design, reporting, interpretation, and academic style that limit its scientific rigor and overall contribution. Below, I detail several specific concerns about the manuscript:
- While the study is described as a pilot, the final sample of 12 participants (6 per group) is very small. Even both control and experimental group has different genders also not equal. Additionally, I would like to note that in some places the sample presented not same (e.g. lines 35 and 126). Dropout of the subjects not explained clearly.
- The power calculation is mentioned but lacks full transparency (no effect size assumptions are presented). With such a small cohort, even moderate statistical significance should be interpreted with extreme caution. The use of inferential statistics (t-tests, ANOVA) is questionable here.
- The worsening of some parameters in the control group (e.g., FM and DLM) is presented as further evidence of intervention efficacy. However, this may reflect measurement noise, lifestyle fluctuations, or seasonal influences. The design lacks blinding or control for confounders.
- A 5-week intervention is very short for eliciting measurable and stable changes in body composition, particularly in fat mass (FM), body fat percentage (BF%), or lean mass. The authors report statistically significant improvements in body composition metrics (e.g., BF%, FM, BMI), yet the brevity of the program makes these changes suspect. Bioimpedance analysis is especially sensitive to fluid shifts, which can distort results in the short term. This concern is especially important when evaluating short-term body composition changes in female participants due to physiological factors unique to women that can significantly influence bioimpedance and fat mass readings.
- The study uses self-reported physical activity (IPAQ) as a key outcome to validate increased PA levels. However, this instrument is known to have poor sensitivity to short-term changes and tends to overestimate activity levels.
- While the MIIE protocol is described in general, there is some inconsistency between the abstract and methods in terms of terminology (e.g., "1'x1'x7" is not explained clearly to all readers). The recovery and work intensity percentages are also not contextualized (why 70% and 40%?). The experiment length is not scientifically proved and content not enough clearly described (e.g. exercises, repetition etc.).
- Physical activity (PA) level is discussed as a significant outcome in the discussion section (lines 300 – 306), yet it is not explicitly presented or interpreted in the results narrative. Since PA is a key variable in the study’s conclusions, I recommend including a clear summary of pre/post changes, p-values, and effect sizes for PA level in the results section to maintain consistency and transparency in reporting.
- Some data presented in research not clear, e.g. “RecHR increased by 8.5 bpm” is unclear. Recovery heart rate normally decreases post-exercise, what is meant here is perhaps an increase in difference from peak?
- The manuscript requires significant editing for clarity, grammar, and academic tone. Redundant expressions (e.g., “physically inactive university students” repeated several times), inconsistent verb tenses, and awkward phrasing reduce readability.
- Tables are hard to interpret due to formatting and alignment issues (e.g., Table 1). What mean different 4 numbers?
- Some sources are cited repeatedly and appear inflated. Also, citation formatting is inconsistent (e.g., missing proper punctuation, spacing).
- Some used terms or abbreviations are not correct:
- PE – usually in scientific literature expressed as “Physical Education”, but not “Physical Exercise”, especially if it is in some parts used as “PE classes” (Lines 207).
- The use of dry lean mass (DLM) in the manuscript requires clarification. While DLM is a valid BIA-derived metric (lean mass excluding total body water), it is not commonly used in exercise studies, and its physiological meaning may be unclear to readers. I recommend briefly defining the term and explaining its limitations, especially in short interventions and even better to use “lean body mass” – more commonly used in scientific articles.
- Abbreviation “EIIM” used in Material and Methods part not clear (line 228).
- The term "intervallic exercise program" is uncommon in exercise science literature. Consider using "interval exercise program" or "interval training program", which are clearer and more widely accepted in the field.
- Reference list for such kind of article is not appropriate. Cited just 39 references and most of them are not published recently (just 17 of them published after 2020 year). Additionally, I would like to note that reference presented not in accordance with MDPI journals requirements.
The manuscript requires significant editing for clarity, grammar, and academic tone. Redundant expressions (e.g., “physically inactive university students” repeated several times), inconsistent verb tenses, and awkward phrasing reduce readability.
Author Response
Comments 1: While the study is described as a pilot, the final sample of 12 participants (6 per group) is very small. Even both control and experimental group has different genders also not equal. Additionally, I would like to note that in some places the sample presented not same (e.g. lines 35 and 126). Dropout of the subjects not explained clearly.
Response 1: The typographical error was corrected (lines 129-134), and it is emphasized that the study corresponds to a pilot with low participation of volunteer students.
Comments 2: The power calculation is mentioned but lacks full transparency (no effect size assumptions are presented). With such a small cohort, even moderate statistical significance should be interpreted with extreme caution. The use of inferential statistics (t-tests, ANOVA) is questionable here.
Response 2: The sample size calculation information was corrected/removed, and the participant selection protocols in the context of a pilot study were detailed (lines 127-130). This consideration was incorporated into the limitations section (lines 398-399), and we discussed the appropriateness of using inferential tests in the context of a pilot study.
Comments 3: The worsening of some parameters in the control group (e.g., FM and DLM) is presented as further evidence of intervention efficacy. However, this may reflect measurement noise, lifestyle fluctuations, or seasonal influences. The design lacks blinding or control for confounders.
Response 3: We agree with the comment. The interpretation of changes in the control group (FM and FFM) was nuanced. The absence of control for confounding factors was emphasized as a methodological limitation.
Comments 4: A 5-week intervention is very short for eliciting measurable and stable changes in body composition, particularly in fat mass (FM), body fat percentage (BF%), or lean mass. The authors report statistically significant improvements in body composition metrics (e.g., BF%, FM, BMI), yet the brevity of the program makes these changes suspect. Bioimpedance analysis is especially sensitive to fluid shifts, which can distort results in the short term. This concern is especially important when evaluating short-term body composition changes in female participants due to physiological factors unique to women that can significantly influence bioimpedance and fat mass readings.
Response 4: We acknowledge the concern raised. The discussion was reinforced to highlight that a 5-week intervention is short, and that the changes observed in body composition can be explained by physiological and metabolic adaptations in subjects with low physical activity levels, beyond an exclusive effect of increased caloric expenditure (lines 370-375). The limitation of bioimpedance regarding fluid variations—particularly in women—was also recognized, but additional details of the measurement protocol were provided (lines 242-246), which demonstrate mastery of the instrument in terms of measurement techniques and factors considered.
Comments 5: The study uses self-reported physical activity (IPAQ) as a key outcome to validate increased PA levels. However, this instrument is known to have poor sensitivity to short-term changes and tends to overestimate activity levels.
Response 5: The limitation of the IPAQ was recognized; therefore, its mention was removed from the results tables and from the main analyses in the “Discussion” section. It is acknowledged only as an instrument that provided relevant information for sample selection in the inclusion criteria.
Comments 6: While the MIIT protocol is described in general, there is some inconsistency between the abstract and methods in terms of terminology (e.g., "1'x1'x7" is not explained clearly to all readers). The recovery and work intensity percentages are also not contextualized (why 70% and 40%?). The experiment length is not scientifically proved and content not enough clearly described (e.g. exercises, repetition etc.).
Response 6: The inconsistency between the “Abstract” and “Methods” sections was corrected. The “1’x1’x7” structure and the exercise intensity applied were detailed based on ACSM guidelines (lines 177-179; 263-265).
Comments 7: Physical activity (PA) level is discussed as a significant outcome in the discussion section (lines 300 – 306), yet it is not explicitly presented or interpreted in the results narrative. Since PA is a key variable in the study’s conclusions, I recommend including a clear summary of pre/post changes, p-values, and effect sizes for PA level in the results section to maintain consistency and transparency in reporting.
Response 7: PAL was removed as a significant outcome, remaining only as an inclusion criterion.
Comments 8: Some data presented in research not clear, e.g. “RecHR increased by 8.5 bpm” is unclear. Recovery heart rate normally decreases post-exercise, what is meant here is perhaps an increase in difference from peak?
Response 8: The wording of this result was corrected to avoid ambiguity, clarifying that “HRR increased by 8.5 bpm,” which refers to a greater difference between peak heart rate and that recorded after 1 minute of recovery, reflecting better recovery capacity (lines 380-381).
Comments 9: The manuscript requires significant editing for clarity, grammar, and academic tone. Redundant expressions (e.g., “physically inactive university students” repeated several times), inconsistent verb tenses, and awkward phrasing reduce readability.
Response 9: Errors in wording, redundancies, and verb tenses were corrected to improve text quality.
Comments 10: Tables are hard to interpret due to formatting and alignment issues (e.g., Table 1). What mean different 4 numbers?
Response 10: Corrected (lines 298-300).
Comments 11: Some sources are cited repeatedly and appear inflated. Also, citation formatting is inconsistent (e.g., missing proper punctuation, spacing).
Response 11: The reference formatting was reviewed and adjusted according to MDPI style (lines 355-578).
Comments 12: Some used terms or abbreviations are not correct: a) PE – usually in scientific literature expressed as “Physical Education”, but not “Physical Exercise”, especially if it is in some parts used as “PE classes” (Lines 207). b) The use of dry lean mass (DLM) in the manuscript requires clarification. While DLM is a valid BIA-derived metric (lean mass excluding total body water), it is not commonly used in exercise studies, and its physiological meaning may be unclear to readers. I recommend briefly defining the term and explaining its limitations, especially in short interventions and even better to use “lean body mass” – more commonly used in scientific articles. c) Abbreviation “EIIM” used in Material and Methods part not clear (line 228). d) The term "intervallic exercise program" is uncommon in exercise science literature. Consider using "interval exercise program" or "interval training program", which are clearer and more widely accepted in the field.
Response 12:
A) The term “Physical Education” was used only to mention the area of expertise of the professional in charge of the evaluation (line 230).
B) The variable “dry lean mass” was explicitly defined (lines 170-171); the abbreviation was corrected.
C) The concept was corrected, using “moderate interval intensity trainning” throughout the text.
Comments 13: Reference list for such kind of article is not appropriate. Cited just 39 references and most of them are not published recently (just 17 of them published after 2020 year). Additionally, I would like to note that reference presented not in accordance with MDPI journals requirements.
Response 13: Six references published after 2020 were incorporated, and the formatting was corrected to comply with MDPI style (line 492, 495, 512, 551, 557 and 566).

Reviewer 4 Report
Comments and Suggestions for Authors
see attached files

Author Response
Comments 1: Line 30: Traditional physical exercise – What does it mean? Some specific form of physical activity? Please elaborate and change accordingly.
Response 1: The term “traditional physical exercise” was removed (line 34).
Comments 2:
Line 33: “which has been minimally applied in the healthcare field”? Please omit this.
Response 2: corrected: was removed (line 34).
Comments 3:
Line 36: “from a state-funded private university in Maule, Chile”. Please omit this, not relevant information for abstract.
Response 3: Corrected (line 36).
Comments 4:
Line 55: “based on the above”, please omit this.
Response 4: Corrected (line 60).
Comments 5:
Lines 89-93: More info about MIIE, elaborate the use of this specific interval training in the study.
Response 5: Further details on MIIT were provided (lines 101-108).
Comments 6:
Lines 15-19: Rephrase sentences and provide clear aim of the study.
Response 6: Incorporated (112-113).
Comments 7:
Line 97: Hypothesis should be presented here.
Response 7: Incorporated (lines 114-115).
Comments 8:
Line 111: “70% n Working heart rate”? Please rephrase.
Response 8: The concept of “working heart rate” was modified to “target heart rate,” which is more commonly used in scientific articles in this field.
Comments 9:
Line 225: “including at least three hydration sessions: before, during and after exercise”? Please elaborate on this or omit. Why 3 hydrations for a 20 min workout?
Response 9: remove.
Comments 10:
Lines 246-253: Whole section should be moved from Results section to Methods section.
Response 10: Corrected (lines 129-134).
Comments 11:
Line 256: Table 1 is not needed, data are presented in Table 2.
Response 11: After attempting to incorporate all the information from Table 1 into Table 2 and eliminating the former, we decided to keep it, since it contains clear and organized information regarding the number of participants per group, sex, PAL, and the p-value of the independent samples t-test conducted to assess baseline differences between groups.
Comments 12:
Line 263: “PAL (p=0.02)” – why was this info presented? PA levels were assessed with IPAQ test, and of course including the 2 sessions per week would increase PA level?
Response 12: Corrected: this section was removed (Table 2).
Comments 13:
Line 269: “demonstrates…” should be omitted, only results should be presented in the RESULTS section.
Response 13: Corrected.
Comments 14:
Lines 281-287: Entire section should be omitted.
Response 14: Corrected: this paragraph in the discussion was reformulated (lines 347-398).
Comments 15:
Lines 309-312: Please elaborate on this statement.
Response 15: Corrected: this information was removed.
Comments 16:
Lines 344-348: A thorough analysis of these studies should be presented and put in the context of current study findings.
Response 16: Corrected: the discussion was reformulated (lines 347-398).
Comments 17:
Lines 354-357: This sentence is too broad, should be rephrased or deleted.
Response 17: Corrected: this information was removed.
Comments 18:
Lines 374-381: Whole section should be deleted, not related to study findings.
Response 18: Corrected: the study’s conclusions were reformulated (lines 424-429)

Reviewer 5 Report
Comments and Suggestions for Authors
The study submitted for review addresses an important issue – low levels of physical activity and excess weight among students, which is associated with the risk of non-communicable diseases. Given the sedentary lifestyle of students, the proposed method of moderate interval loads can be an effective and affordable solution. The developed program showed positive changes in body composition, low volume and moderate intensity make the program easily applicable in university settings.
Despite the scientific novelty of the study, its practical and theoretical significance, we note several comments and suggestions of a debatable nature. The study only included 12 participants, which reduces the statistical power and generalizability of the results. The short duration of the experimental intervention - 5 weeks, which is too short for long-term conclusions about the effect on body composition. The study records the lack of control over nutrition and other factors, and did not take into account diet, sleep, stress, which can affect body composition. It should also be noted that non-randomized distribution, namely, convenience sampling, can distort the results. As a result, there is no comparison with other types of training, such as aerobic or strength training.
However, the study demonstrates the potential of the method as a convenient and effective method for improving body composition in students. However, due to the small sample size and short period, larger and longer-term studies are needed. The work contributes to the development of low-impact and accessible physical activity programs for young people.
Author Response
Comments 1: Despite the scientific novelty of the study, its practical and theoretical significance, we note several comments and suggestions of a debatable nature. The study only included 12 participants, which reduces the statistical power and generalizability of the results. The short duration of the experimental intervention - 5 weeks, which is too short for long-term conclusions about the effect on body composition. The study records the lack of control over nutrition and other factors, and did not take into account diet, sleep, stress, which can affect body composition. It should also be noted that non-randomized distribution, namely, convenience sampling, can distort the results. As a result, there is no comparison with other types of training, such as aerobic or strength training.
Response 1: We acknowledge the observations regarding the methodological limitations. The points raised were incorporated into the limitations section (lines 398-404), and we explained some of the decisions made.
Comments 2: However, the study demonstrates the potential of the method as a convenient and effective method for improving body composition in students. However, due to the small sample size and short period, larger and longer-term studies are needed. The work contributes to the development of low-impact and accessible physical activity programs for young people.
Response 2: We appreciate the reviewer’s conclusion regarding the potential of this study and its applicability. We agree that longer trials with larger samples are required, and this has been incorporated into the “Guidelines for Future Research" section (lines 412-422) .

Round 2
Reviewer 1 Report
Comments and Suggestions for Authors
The manuscript has been considerably improved and meets the standards required for publication.
Author Response
Comments 1: The manuscript has been considerably improved and meets the standards required for publication.
Response: I appreciate your suggestions and comments to improve the manuscript
Reviewer 2 Report
Comments and Suggestions for Authors
Dear authors,
Thank you for addressing my comments.
Best regards,
Author Response
Comments: Dear authors, Thank you for addressing my comments. Best regards.
Response: Dear reviewer, we are grateful for your collaboration, which made it possible to improve our manuscript
Reviewer 3 Report
Comments and Suggestions for Authors
The authors have improved their article according to my comments, although there are still some places for improvement:
- Authors have tried to explain BIA usage limitations, but in especially female sample not explained clearly (e.g. especially cycle-phase control) additionally how was controlled fluid-related BIA outputs.
- In Tables usually data must be presented with dots, but not with commas (e.g. table 1 and table 2).
- Not necessary to use abbreviations if it is used just once in the article (e.g. BPF in lines 230).
The authors have improved their article according to my comments, although there are still some places for improvement:
- Authors have tried to explain BIA usage limitations, but in especially female sample not explained clearly (e.g. especially cycle-phase control) additionally how was controlled fluid-related BIA outputs.
- In Tables usually data must be presented with dots, but not with commas (e.g. table 1 and table 2).
- Not necessary to use abbreviations if it is used just once in the article (e.g. BPF in lines 230).
Author Response
Comment 1: Authors have tried to explain BIA usage limitations, but in especially female sample not explained clearly (e.g. especially cycle-phase control) additionally how was controlled fluid-related BIA outputs.
Response 1: We have added important information regarding the control of BIA measurements in female participants (lines 250–253). In addition, some related exclusion criteria were incorporated (lines 157–159).
Comment 2: In Tables usually data must be presented with dots, but not with commas (e.g. table 1 and table 2).
Response 2: Corrected (Table 1 and Table 2).
Comment 3: Not necessary to use abbreviations if it is used just once in the article (e.g. BPF in lines 229).
Response 3: Abbreviations of concepts mentioned fewer than five times in the text were removed: NCDs (lines 31, 55, 68, and 349), WHO (line 78), WMA (line 453), and BPF (line 235).
General comment: In general, we have made modifications to the English wording of the entire text.
Additional information was included in the "Procedures" section to better explain the sequence of the intervention.
We sincerely thank you in advance for your valuable collaboration in improving our project.
